# WHEN NAMES DISAPPEAR: REVEALING WHAT LLMS ACTUALLY UNDERSTAND ABOUT CODE

## ABSTRACT

Large Language Models (LLMs) achieve strong results on code tasks, but how they derive program meaning remains unclear. We argue that code communicates through two channels: structural semantics, which define formal behavior, and human-interpretable naming, which conveys intent. Removing the naming channel severely degrades intent-level tasks such as summarization, where models regress to line-by-line descriptions. Surprisingly, we also observe consistent reductions on execution tasks that should depend only on structure, revealing that current benchmarks reward memorization of naming patterns rather than genuine semantic reasoning. To disentangle these effects, we introduce a suite of semantics-preserving obfuscations and show that they expose identifier leakage across both summarization and execution. Building on these insights, we release CLASSEVAL-OBF, an obfuscation-enhanced benchmark that systematically suppresses naming cues while preserving behavior. Our results demonstrate that CLASSEVAL-OBF reduces inflated performance gaps, weakens memorization shortcuts, and provides a more reliable basis for assessing LLMs' code understanding and generalization.

## 1 INTRODUCTION

Large language models (LLMs) now achieve striking results across code intelligence—program synthesis, repair, summarization, and test generation. Yet how these models derive meaning from source code remains unclear. We posit that code communicates through *two channels*: a **structural/semantic** channel (syntax, control/data flow, execution behavior) and a **human–naturalness** channel (identifier names, docstrings, and other linguistic signals). If an LLM truly understands a program's intent, its behavior should remain stable when human-interpretable names are perturbed while semantics stay fixed; conversely, strong performance drops would indicate an overreliance on surface cues rather than semantic reasoning.

To probe this question, we move beyond a single perturbation (e.g., $\alpha$-renaming) and introduce a *suite of semantics-preserving obfuscations* that progressively weaken the naturalness channel while preserving executable behavior. Our suite includes 1) simple structural renaming (alpha-renaming): replace identifiers with role-preserving placeholders; 2) Ambiguous identifiers: replace identifiers with visually ambiguous tokens; 3) Cross-domain terms: substitute identifiers with terms from unrelated fields to break semantic cues in the application domains; and 4) Misleading semantics: assign names that imply incorrect behaviors.

These transformations define a rename–obfuscation spectrum from minimally disruptive (synonyms) to strongly disruptive (opaque tokens), enabling a graded analysis of robustness. We then evaluate models on complementary task families that stress distinct facets of "understanding": **intent summarization** (what/why) and **execution/IO prediction**. We stratify data across *intent-rich, real-world* code (where names and headers carry domain semantics) and *algorithmic, competitive-programming* code (where identifiers are already minimal and structure is highly diagnostic).

This design yields three core observations. First, on intent-rich code, *class- and method-level summarization degrades sharply* under strong obfuscation (especially entity-level renaming), often collapsing into line-by-line narration. Second, on competitive-programming solutions, *summaries remain intent-faithful* under obfuscation, consistent with the view that structure alone is sufficient when algorithmic patterns are canonical and naming is sparse. Third, and most surprisingly,

even *execution-oriented* tasks—ostensibly dependent only on program semantics—show non-trivial drops after obfuscation, suggesting that existing benchmarks permit shortcuts in which identifiers act as retrieval cues for memorized patterns rather than triggering genuine reasoning. Our additional stress tests with input augmentation indicate that removing names reduces these retrieval effects, narrowing the gap between the *appearance* of step-wise reasoning and *actual* generalization.

Our study contributes a principled framework for disentangling the two channels of code understanding and for diagnosing where models rely on names versus structure. Methodologically, we pair a capture-avoiding obfuscation harness with *semantics-invariance checks* (execution correctness) and report *human-aligned intent metrics* via code summarization. Finally, we also release CLASSEVAL-OBF, an obfuscation-enhanced variant of class- and function-level benchmarks to promote evaluations that are less susceptible to naming leakage, available at `https://classeval-obf.site`

## 2 RELATED WORK

**Structure-aware and execution-grounded modeling.** Modern code models increasingly fuse *structural* program information and *execution* signals. Guo et al. (2021) pre-train GraphCode-BERT with data-flow edges and variable alignment, showing that incorporating semantic structure improves downstream code understanding tasks (search, clone detection, translation, refinement). Tehrani Jamsaz et al. (2023) introduce PerfoGraph, a program-graph representation that injects numerical and aggregate-structure information to better capture program behavior. Li et al. (2023) propose SANTA, a structure-aware dense retrieval model that aligns structured and unstructured data via contrastive pretraining and entity-masked prediction. Wu et al. (2024) introduce a plug-and-play method that leverages AST-based structure loss during fine-tuning to enhance pretrained code LLMs, particularly under limited training data. Le et al. (2022) propose CodeRL, which grounds generation in *unit-test execution feedback* via reinforcement learning to optimize functional correctness. Together, these lines suggest that beyond token sequences, structure- and execution-aware learning is essential to robust code understanding—an assumption our study probes from a complementary angle by isolating the role of human-interpretable naming.

**Identifier naming, robustness, and obfuscation.** A growing body of work shows that model predictions can be sensitive to identifier naming and obfuscation. Gao et al. (2023) demonstrate substantial performance drops from simple identifier renaming and propose Cream, a counterfactual framework to separate helpful from misleading identifier signals across code understanding tasks. In a related direction, Yang et al. (2022) study *natural* adversarial attacks on code models (including variable renaming) and document brittleness to lexical changes that preserve semantics. Lam et al. (2025) introduce a unified benchmark that applies logic-preserving perturbations—ranging from systematic renaming and conditional rewrites to misleading comments and garbage code—to reveal overreliance on natural-language cues and reasoning collapse in LLMs. Our obfuscating renaming experiments build directly on these observations: by holding program semantics fixed while removing human-interpretable names, we quantify how much intent-level understanding depends on the "naturalness" channel versus program semantics.

**Benchmarks, evaluation scope, and SE perspectives.** Benchmarks for evaluating large language models (LLMs) on code tasks have evolved from short snippet assessments to more realistic software engineering scenarios. Early benchmarks such as HUMANEVAL (Chen et al., 2021), MBPP (Austin et al., 2021), EVALPLUS (Liu et al., 2023), CodeXGlue (Lu et al., 2021), CodeApex (Fu et al., 2023), CodeMMLU (Nguyen et al., 2025), and BigCodeBench (Zhuo et al., 2024) primarily target function- or snippet-level tasks. In contrast, LiveCodeBench (Jain et al., 2025) and Code-Contests (Li et al., 2022) emphasize competitive programming. More recent efforts—including ClassEval (Du et al., 2024a), SWE-Bench (Jimenez et al., 2024), SWE-Gym (Pan et al., 2025), SWE-Bench-live (Zhang et al., 2025), Defects4J (Just et al., 2014), and BugsInPy (Widyasari et al., 2020)—extend evaluation to class- and repository-level settings, revealing capability gaps not captured by function-only benchmarks. Parallel research has advanced execution-grounded evaluation. CruxEval (Gu et al., 2024) formalized input–output prediction, while LiveCodeBench (Jain et al., 2025) expanded this to human-written solutions. Benchmarks such as REval (Chen et al., 2025) and CACP (Hooda et al., 2024) identified concept-level reasoning failures, and execution-based frameworks including LEVER (Ni et al., 2023), CodeScore (Dong et al., 2025), and XCODEEVAL (Khan et al., 2024) refined evaluation methodology. Robustness has also been studied through semantic-

preserving transformations, such as ReCode (Wang et al., 2022). From the synthesis perspective, reinforcement learning and feedback-based methods emphasize behavior-grounded evaluation (Le et al., 2022; Pham et al., 2025; Yang et al., 2025; Gehring et al., 2024). Complementary surveys and empirical studies further highlight concerns about robustness and validity in developer-facing tasks (Gao et al., 2023; Lam et al., 2025). Our work complements these trends by proposing $\alpha$-renaming as a *controlled* stress test that distinguishes intent-level summarization from behavior-level execution, and by recommending reporting protocols (e.g., pre/post renaming deltas and uncertainty).

# 3 PRELIMINARY EXPERIMENT AND MOTIVATION

Unlike natural languages, programming languages allow developers to write programs that instruct machines to perform specific tasks. Source code conveys program meaning through two complementary channels: *program semantics*, which captures the formal behavior of a program via program constructs, and *human-interpretable naming* (naturalness channel Hindle et al. (2012)), where identifiers and comments convey *program intent* (we focus on identifiers since not all programs have comments). While program constructs may be sufficient to recover canonical algorithms, intent-level understanding in real-world code often relies on naming. If LLMs truly reason and have full comprehension on source code, removing the naming channel should primarily affect tasks that require intent inference, such as summarization, while leaving execution-oriented tasks largely intact.

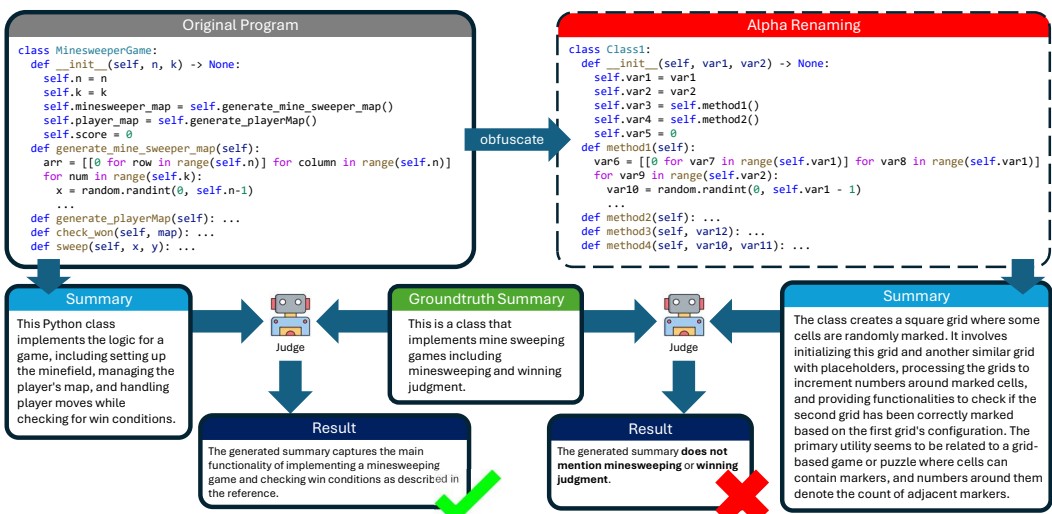

Figure 1: **Names as semantic anchors for summarization.** LLM produces an intent-level summary with the original identifiers (left), but collapses to line-by-line narration after name-only obfuscation (right), despite identical structure and behavior.

As a first step, we focus on **code summarization** because it explicitly targets program intent rather than detail-level program constructs. To isolate the role of the human-interpretable channel, we obfuscated variable and method names while preserving the program's structure. Figure 1 illustrates this using a Python CLASSEVAL dataset Du et al. (2024b) example implementing a Minesweeper game. In the original version, names such as MinesweeperGame, sweep, check_won, and coordinates x, y serve as semantic anchors that strongly hint at the domain and gameplay logic. After obfuscation, these anchors were removed, though the control and data flows remained unchanged.

When prompted to generate a summary, the model's behavior diverged sharply. On the original code, the GPT-4o correctly identified a *Minesweeper game*, describing initialization, gameplay logic, and winning conditions. On the obfuscated version, it failed to capture the concept, instead degenerating into line-by-line descriptions of grid updates and neighbor increments without naming the task or intent. This contrast suggests that identifiers are disproportionately influential for summarization, anchoring structural patterns to high-level human concepts.

This example illustrates that identifiers serve as critical semantic anchors for summarization, a task that depends on program intent. Once obfuscated, the model loses these anchors and collapses

into surface-level, line-by-line narration despite identical structure and behavior. While compelling, this example alone is anecdotal. To rigorously evaluate the role of naming in intent-level under-standing, we next conduct systematic experiments across datasets and models in Section 4, includ-ing both intent-rich code (CLASSEVAL) and competitive programming tasks (LIVECODEBENCH) where naming plays a smaller role due to the nature of challenges in algorithms and data structures.

From the first example, we observed that removing identifiers strongly impairs LLMs on summariza-tion, a task that requires recovering program intent through the naturalness channel. We now shift to a different task: **code execution prediction**. Unlike summarization, this task mainly requires rea-soning about program semantics and line-by-line behavior. Since our obfuscation preserves syntax, control flow, and data flow, model performance should in principle remain stable if predictions truly rely on program structure alone.

Table 1: **Motivating experiment on output prediction tasks.** We compare model performance (Pass@1 and Pass@3) on CLASSEVAL and LIVECODEBENCH with original vs. obfuscated code.

| Dataset | Pass@1 | | Pass@3 | |
|---|---|---|---|---|
| | Original | Obfuscated | Original | Obfuscated |
| ClassEval | 85.7 | 76.1 | 89.2 | 83.3 |
| LiveCodeBench | 85.4 | 71.2 | 97.9 | 80.1 |

Table 1 summarizes the results. Contrary to expectation, program execution prediction performance degrades in both datasets after identifiers were removed. On CLASSEVAL, Pass@1 drops from 85.7 to 76.1 and Pass@3 from 89.2 to 83.3. Even more striking, on LIVECODEBENCH—where naming conventions are sparse and implementations emphasize algorithmic structure—we still observed sharp declines: Pass@1 falls from 85.4 to 71.2, and Pass@3 from 97.9 to 80.1.

These preliminary findings indicate that LLMs exploit statistical correlations between identifiers and functionality even for tasks that should be un-affected by renaming. This contradicts the common assumption that execution benchmarks cleanly capture structural reasoning about program behavior. Why do identifiers matter so much in execution prediction? We address this puzzle in Section 5, where we analyze obfuscation strategies, disentangle memorization from genuine reasoning, and study how naming biases shape execution performance. Building on these insights, we then propose in Section 6 new obfuscation-based benchmarks that more reliably measure code understanding by separating structural semantics from memorization effects.

## 4 PROGRAM INTENT CHANNEL: CODE SUMMARIZATION

### 4.1 EXPERIMENTAL SETUP

We design our experiment to evaluate models' ability to capture program intent rigorously.

**Datasets:** We select two diverse corpora that complement each other in focus and complexity. The CLASSEVAL benchmark targets class- and method-level summarization, focusing on developers' intent beyond code constructs. In contrast, LIVECODEBENCH provides algorithmically oriented competitive programming tasks, which challenge models to capture procedural structure rather than high-level intent. This combination allows us to probe the impacts of both naturalness and program semantic channels on LLM code comprehension.

**Models:** We evaluate a set of advanced LLMs that span a range of scales and architectures: GPT-4o, Qwen3-Coder 480B, DeepSeek V3 0324, and Llama 4 Maverick. By including both top-tier and mid-scale models, we examine how model capacity influences understanding of program intent and structural fidelity.

**Metrics:** For CLASSEVAL, we report class-level and method-level scores, reflecting both holistic and fine-grained comprehension. For LIVECODEBENCH, we compute method-level accuracy that measures alignment with algorithmic correctness. In all cases, scores are averaged across samples to provide consistent, comparable metrics.

**LLM-as-a-judge:** Summaries are evaluated using rubric-based scoring derived from the BigGen-Bench framework Kim et al., which emphasizes semantic fidelity and comprehensive understand-

ing. We employ GPT-4o as the evaluation model, chosen for its stability and self-consistency across repeated judgments. Each summary is rated along five dimensions—intent capture, behavioral coverage, algorithmic adequacy, faithfulness, and clarity—on a 1–5 scale, then normalized to 0–100. This setup ensures that metrics reflect semantic reasoning rather than superficial lexical overlap.

**Pipeline:** For each sample, the target model generates a summary, which is then scored by the GPT-4o judge according to the rubrics. Results are aggregated across obfuscation strategies, enabling direct comparison between original and obfuscated variants. This pipeline provides a rigorous, semantically grounded measure of summarization performance.

## 4.2 EXPERIMENTAL RESULT

Table 2: **Summarization performance under name obfuscation.** We report class-level and method-level accuracy on CLASSEVAL, and method-level accuracy on LIVECODEBENCH.

| Model | ClassEval | | | | LiveCodeBench | |
| | Class Acc. | | Method Acc. | | Method Acc. | |
| | Original | Obfuscated | Original | Obfuscated | Original | Obfuscated |
|---|---|---|---|---|---|---|
| GPT-4o | **87.3** | 58.7 | **65.9** | 44.6 | **73.4** | 70.1 |
| Qwen3–Coder 480B | **87.2** | 72.1 | **55.9** | 49.6 | **87.7** | 82.1 |
| DeepSeek V3 | **87.7** | 76.7 | **56.2** | 51.9 | **83.6** | 78.4 |
| Llama 4 Maverick | **86.2** | 66.4 | **56.0** | 48.1 | 77.5 | **78.2** |

**Analysis.** Summarization performance diverges sharply across the two benchmarks. On CLASSE-VAL, class-level accuracy collapses after obfuscation, with drops ranging from 11 points (DeepSeek V3: 87.7 → 76.7) to almost 29 points (GPT-4o: 87.3 → 58.7). Method-level summarization is also affected, though to a lesser extent, with declines averaging around 9–12 points across models. These results confirm that summarization relies heavily on the naturalness channel—especially entity names—at the class level, where names and headers carry substantial intent.

In contrast, performance on LIVECODEBENCH remains remarkably stable. The average reduction is below three points, and in one case (Llama 4 Maverick) obfuscation even produces a slight improvement (77.5 → 80.2). This resilience stems from the nature of competitive programming tasks: identifiers are typically generic (a, b, n), and the program's purpose is communicated through structural and algorithmic cues rather than naming. To quantify this difference, we compare the lengths of variable names in the two datasets. As shown in Figure 2, CLASSEVAL exhibits substantially longer and more descriptive identifiers (median length ≈ 8, with many exceeding 15 characters), whereas LIVECODEBENCH identifiers are extremely short (median length ≈ 2), often restricted to one-letter variables. This statistical contrast supports the observation that models trained on competitive programming data cannot rely heavily on naming cues, explaining their robustness under obfuscation.

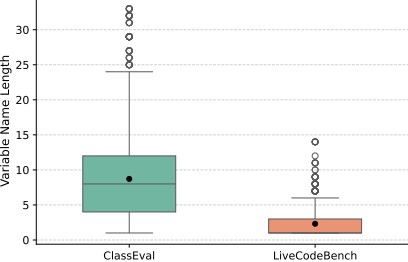

Figure 2: Distribution of variable name lengths in CLASSEVAL and LIVECODEBENCH.

Model scale and training diversity also matter. Qwen3–Coder 480B and DeepSeek V3 both maintain higher robustness, dropping *less* than smaller counterparts. However, GPT-4o, while very strong on the original code, shows the *steepest decline* once identifiers are removed—indicating that its advantage is partly tied to exploiting semantic-rich naming rather than purely structural reasoning.

Qualitative inspection reinforces this view. On CLASSEVAL, the summaries of original code typically capture high-level intent (e.g., "implements caching mechanism"), whereas obfuscated versions often degrade into verbose, line-by-line narrations of control flow. This discrepancy highlights how strongly models depend on naming cues when inferring intent-level meaning. In contrast, on LIVECODEBENCH, the difference is minimal: summaries of both original and obfuscated programs remain essentially intact. For instance, the function below—regardless of whether variables are renamed—is consistently summarized as *"the code converts a given string into a palindrome by modifying characters symmetrically."* Unlike in CLASSEVAL, there is no regression into line-by-line paraphrasing, confirming that when summarizing the competitive programming code, a model tends to rely more on algorithmic structure than on naming cues.

```python
def makeSmallestPalindrome(s: str) -> str:
    s = list(s)
    n = len(s)
    for i in range(n):
        c = min(s[i], s[n - 1 - i])
        s[i] = c
        s[n - 1 - i] = c
    return "".join(s)
```

*Key Findings.* Name-obfuscation evaluation indicates that benchmarks like CLASSEVAL better measure model performance on intent-understanding tasks, whereas low-naturalness settings (e.g., competitive-programming code in LIVECODEBENCH) may under-challenge LLMs and mask limitations. Accordingly, incorporating identifier obfuscation into benchmark design is essential for robust assessment of program intent-level tasks such as code summarization or code review.

## 5 PROGRAM SEMANTIC CHANNEL: CODE EXECUTION PREDICTION

### 5.1 EXPERIMENTAL SETUP

We establish the following experimental setup to rigorously assess the ability of LLMs to perform code execution reasoning under varying naming conventions.

**Datasets.** We evaluate on two complementary benchmarks as in the experiment in Section 4. To focus on non-trivial execution behavior, we filter CLASSEVAL to retain only samples with Cyclomatic Complexity $\geq 15$, and LIVECODEBENCH to retain only samples with Cyclomatic Complexity $\geq 6$. After filtering, the evaluation set comprises 37 CLASSEVAL instances and 96 LIVECODEBENCH solutions. This high-complexity selection ensures the tasks demand substantial code reasoning and emphasize any meaningful effects of obfuscation.

**Models.** We evaluate a representative set of state-of-the-art LLMs, including GPT-4o, Qwen3-Coder 480B, and DeepSeek V3, ensuring that we include frontier-scale and mid-scale models.

**Metrics.** For each sample, we ran 5 times and calculate the average *Pass@1* and *Pass@3* values.

**Obfuscation strategies.** To probe how naming affects LLM reasoning while preserving program behavior, we apply four deterministic, name-only obfuscation strategies:

- **Simple structural renaming (alpha-renaming)**: replace identifiers with role-preserving placeholders such as `class1, class2, method1, var1, var2, ...`.

- **Ambiguous identifiers**: replace identifiers with visually ambiguous tokens (examples: `llllIII, Il1lI1lllIlI`).

- **Cross-domain terms**: substitute identifiers with terms from unrelated fields (e.g., medical: `adrenaline_fd, glucagon_d6`) to break semantic cues in the application domains.

- **Misleading semantics**: assign names that imply incorrect behaviors (e.g., a summing function named `compute_max`).

All obfuscations, as illustrated in Figure 3, are deterministic, reproducible, and change only the naturalness channel of the code and do not change program semantics.

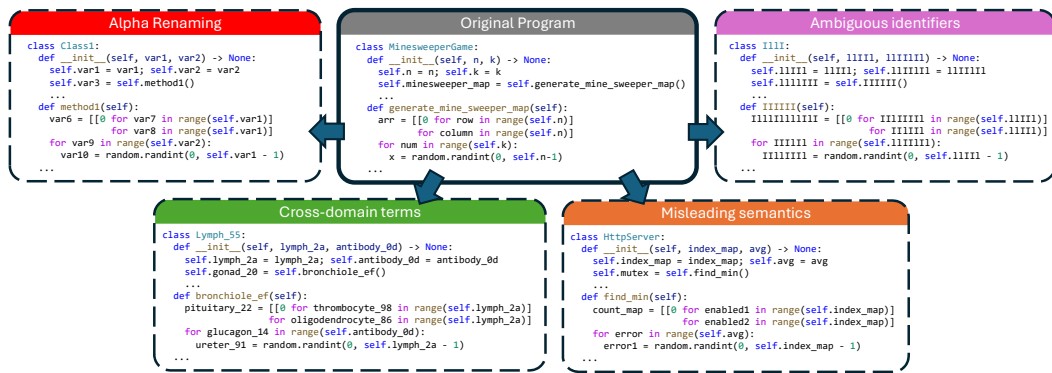

Figure 3: Illustration of the four obfuscation strategies applied in our study.

Table 3: Execution prediction performance across original code and four obfuscation strategies. Values in green equal or exceed the model's *Orig* for that dataset; red are lower.

| | ClassEval | | | | | | | | | |
|---|---|---|---|---|---|---|---|---|---|---|
| Model | Orig | | Alpha | | Ambiguity | | CrossDomain | | Misleading | |
| | Pass@1 | Pass@3 | Pass@1 | Pass@3 | Pass@1 | Pass@3 | Pass@1 | Pass@3 | Pass@1 | Pass@3 |
| GPT-4o | **76.6** | **84.6** | 70.2 | 75.4 | 74.7 | 81.0 | 75.1 | 84.6 | 71.1 | 76.5 |
| Qwen3–Coder 480B | **92.6** | **98.7** | 86.4 | 90.0 | 87.3 | 90.9 | 88.2 | 92.7 | 83.7 | 88.2 |
| DeepSeek V3 | **90.0** | **92.7** | 85.5 | 92.5 | 69.3 | 73.8 | 89.8 | 91.8 | 88.8 | 91.6 |
| Llama 4 Maverick | **81.1** | **85.5** | 83.4 | 86.1 | 72.0 | 76.5 | 79.9 | 85.3 | 80.8 | 86.2 |

| | LiveCodeBench | | | | | | | | | |
|---|---|---|---|---|---|---|---|---|---|---|
| Model | Orig | | Alpha | | Ambiguity | | CrossDomain | | Misleading | |
| | Pass@1 | Pass@3 | Pass@1 | Pass@3 | Pass@1 | Pass@3 | Pass@1 | Pass@3 | Pass@1 | Pass@3 |
| GPT-4o | **82.9** | **95.9** | 75.5 | 90.4 | 68.7 | 78.9 | 72.1 | 85.7 | 82.3 | 92.5 |
| Qwen3–Coder 480B | **99.3** | **100** | 94.5 | 98.6 | 88.4 | 96.5 | 89.7 | 93.1 | 93.8 | 97.9 |
| DeepSeek V3 | **99.3** | **100** | 98.6 | 99.3 | 91.8 | 95.2 | 98.6 | 100 | 97.2 | 97.9 |
| Llama 4 Maverick | **80.2** | **90.4** | 65.3 | 78.9 | 56.4 | 69.3 | 63.9 | 76.8 | 75.5 | 88.4 |

## 5.2 EXPERIMENTAL RESULTS

**Analysis.** Table 3 provides the central quantitative evidence of our study. Intuitively, one would expect that the name obfuscation does not affect a model's reasoning on the execution semantics of the obfuscated code. Surprisingly, across both benchmarks, name obfuscation consistently reduces accuracy, though the magnitude varies by model and dataset. On CLASSEVAL, the degradation is moderate but widespread. For instance, GPT-4o drops from 76.6% to 70.2% Pass@1 under *Alpha*, and DeepSeek V3 falls drastically from 90.0% to 69.3% under *Ambiguity*. These results reveal that even large, high-performing models are vulnerable once surface cues are altered. Still, a few cases conform to intuition: Llama 4 Maverick surpasses its original performance under *Alpha*. Such cases suggest that when the reasoning pathway is unaffected, identifier names function only as replaceable placeholders. Figure 4 is an illustration of such cases where the model still reasons well on heavily name-obfuscated code.

The picture is more dramatic in LIVECODEBENCH. Nearly all models suffer double-digit drops, with reductions exceeding 20–30% in several obfuscation settings. For example, Llama 4 Maverick falls from 80.2% to just 56.4%. Even GPT-4o and Qwen3-Coder 480B, which remain strong overall, show consistent declines. Only isolated cases (*CrossDomain* for DeepSeek V3) yield equal or higher scores, and these are rare exceptions rather than the dominant trend.

If obfuscation only renames variables without changing program semantics, the table is expected to be dominated with green cells (equal or better than original). Instead, red cells are overwhelmingly prevalent, showing that accuracy is tightly linked to naming cues. Figure 4 shows that reasoning remains stable and unaffected by renaming, yet the aggregate outcomes collapse under obfuscation.

This tension—stable per-step reasoning but degraded aggregate accuracy—raises critical questions about the reliability of LLM performance. Specifically, if the model can reason line-by-line as

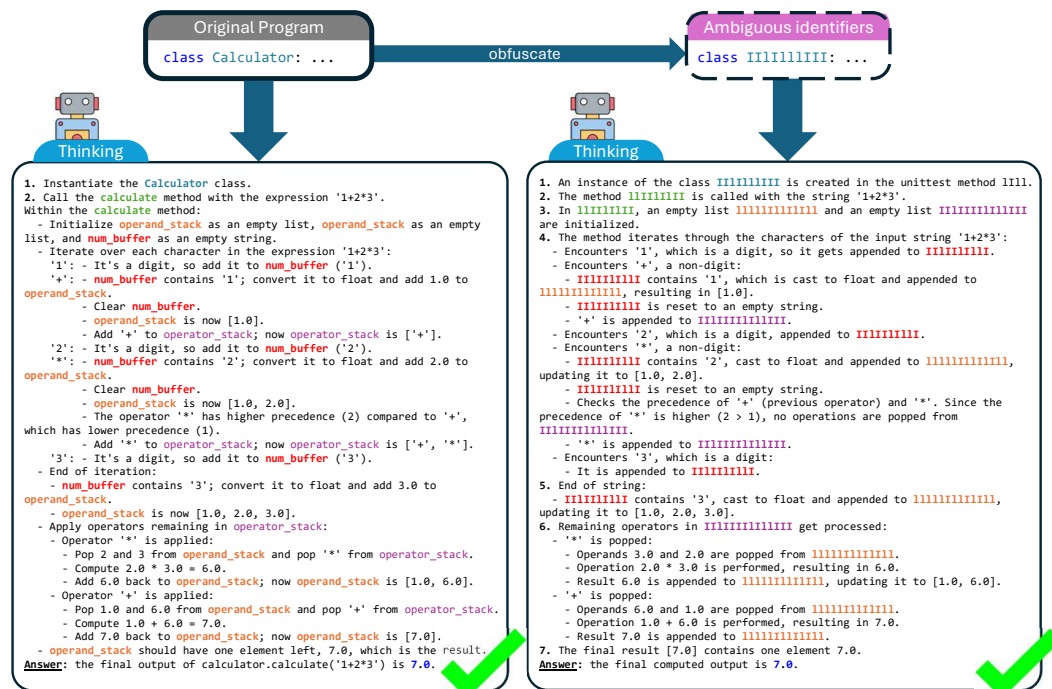

Figure 4: Qualitative example: GPT–4o's step-by-step reasoning on the original (left) and an *Ambiguous identifiers* obfuscation (right) of the same program yields the same final correct result (7.0).

illustrated in Figure 4, why does name obfuscation still reduce final accuracy? One possibility is that LLMs exploit surface-level cues in the original code. Rather than genuinely reasoning in all cases, the model may occasionally rely on memorized associations keyed by familiar identifiers. Under this interpretation, obfuscation weakens these cues, revealing a gap between the appearance of reasoning and actual generalization.

To investigate this hypothesis, we conduct a *memorization stress test*, augmenting inputs and measuring cases where the model reproduces outputs from the training distribution. If identifier names act as "access keys" for memorization, obfuscation should disrupt such matches.

**Memory effect experiment.** Building on the hypothesis raised in the qualitative analysis, we conducted a dedicated experiment to probe whether LLMs rely on memorization rather than genuine input-driven reasoning.

*Setup.* We first used GPT-4o to generate new input values for each sample in the datasets, ensuring the outputs differ from those in the original datasets. We then filtered out cases with small finite output domains (e.g., `True`/`False`) to minimize chance overlap. Model predictions are compared against both the old dataset outputs and the new ground-truth outputs. The key question is whether the model adapts to unseen inputs or instead reproduces memorized outputs from prior exposure.

*Observation.* Table 4 shows that on the original (non-obfuscated) code, models occasionally reproduce the *old* outputs instead of the correct new ones, with GPT-4o and Llama 4 Maverick showing the strongest signs of this effect on LIVECODEBENCH. The probability of such matches occurring by chance is negligible, confirming that models sometimes fall back on memorized associations. Under obfuscation, however, this effect sharply decreases—often to zero—supporting the view that variable names act as retrieval cues or "keys" for accessing memorized code–output patterns.

*Findings.* After these analyses, we highlight the following key insights:

- **Identifier leakage.** Variable names anchor memorized code–output pairs in training data; obfuscation disrupts this shortcut and pushes the model toward execution reasoning.

- **Benchmark inflation.** Existing execution benchmarks risk overestimating reasoning ability, since identifier leakage enables partial memorization to masquerade as generalization.

Table 4: Memorization check: #Samples where prediction equals the *old dataset output* across new augmented inputs.

| Model | ClassEval | | | LiveCodeBench | | |
|---|---|---|---|---|---|---|
| | Orig | Ambiguity | Misleading | Orig | Ambiguity | Misleading |
| GPT-4o | **1** | 0 | 0 | **13** | 5 | 2 |
| Qwen3-Coder 480B | **1** | 0 | 0 | **2** | 0 | 0 |
| DeepSeek V3 0324 | **2** | 0 | 0 | 0 | 0 | 0 |
| Llama 4 Maverick | 0 | 0 | 0 | **9** | 5 | 3 |

- **Toward robust evaluation.** Reliable benchmarks for code execution understanding evaluation should integrate (i) augmented datasets with novel input–output pairs, and (ii) systematic obfuscation of identifiers. These measures jointly reduce memory-based shortcuts and provide a clearer assessment of semantic reasoning ability.

## 6    CLASSEVAL-OBF: A RELIABLE DATASET FOR EXECUTION PREDICTION

Building on these findings, we release **CLASSEVAL-OBF**, an obfuscated extension of CLASSEVAL designed to mitigate identifier leakage and provide a more faithful measure of execution reasoning. In this dataset, all identifiers are systematically renamed using four complementary strategies (*alpha-renaming, ambiguous identifiers, cross-domain substitutions, misleading semantics*), ensuring that program behavior is preserved while surface-level cues are removed or distorted.

Figure 5: Execution prediction performance on original vs. obfuscated CLASSEVAL (high-complexity subset)

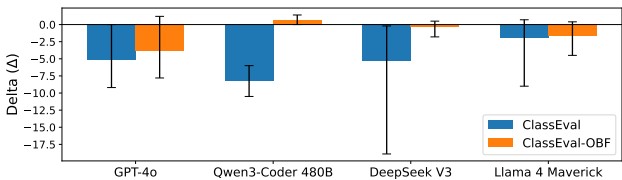

**Evaluation.**    We re-run execution prediction experiments on the high-complexity subset, comparing original CLASSEVAL with its obfuscated variants. Figure 5 reports the performance delta ($\Delta$) between original and obfuscated code across the four obfuscation strategies, summarized as minimum, maximum, and average values. Smaller deltas indicate higher robustness, as models maintain accuracy even when identifier names are removed.

Across models, CLASSEVAL-OBF consistently reduces the magnitude of performance drops compared to the original CLASSEVAL, with most deltas confined within 3–7%. In several cases, the delta is nearly zero or even slightly positive, suggesting that obfuscation can prevent models from overfitting to naming cues. This pattern confirms that CLASSEVAL-OBF mitigates identifier leakage and provides a more stable, semantics-grounded benchmark for execution reasoning.

## 7    CONCLUSION

We set out to disentangle how LLMs "understand" code by separating a structural/semantic channel from a human–naturalness channel and evaluating models under a suite of semantics-preserving obfuscations that progressively suppress names and prose while keeping behavior intact. Across intent summarization and execution prediction—and across real-world and competitive-programming settings—we observed a consistent, graded pattern: intent-level performance degrades sharply as naturalness is removed, while behavior-level metrics remain largely invariant except where names are semantically active. These results provide converging empirical support for a two-channel account of code understanding and motivate evaluation practices that report pre/post obfuscation deltas alongside human-aligned intent metrics. We release our obfuscation harness and protocols to encourage benchmarks that reward true semantic reasoning over surface cues and to catalyze progress toward models that capture program intent, not just its narration.

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

## A    APPENDIX

### A.1    PROMPT FOR CODE SUMMARIZATION

```
You are a helpful assistant for code understanding.
Summarize the following Python class at two levels:
1. The overall class.
2. Each method individually.

Return your answer strictly in JSON format:
{
  "class_summary": "...",
  "methods": [
    {"method_name": "method1", "method_summary": "..."},
    {"method_name": "method2", "method_summary": "..."}
  ]
}

Here is the class code:

```python
{code}
```
```

Figure 6: Prompt used for ClassEval class and method-level summarization.

```
You are a helpful assistant for code understanding.
Summarize the main functionality of this function.

```python
{code}
```
```

Figure 7: Prompt used for LiveCodeBench function-level summarization.

```
You are an expert code reviewer. Evaluate how well a generated
summary matches the gold reference.

### Scoring Rubric
- 5/5 (Excellent): Semantically equivalent to reference; covers the
main purpose accurately; concise; no major omissions or errors.
- 4/5 (Good): Mostly correct; captures main idea but misses a minor
detail or is slightly imprecise.
- 3/5 (Fair): Partially correct; captures some intent but misses
important elements or includes minor inaccuracies.
- 2/5 (Poor): Only loosely related; major inaccuracies or misses
most key functionality.
- 1/5 (Very poor): Irrelevant or completely wrong.

### Examples
Reference: "Sorts a list of numbers in ascending order."
Generated: "Reverses a list."
Score: 1/5
Justification: Opposite functionality; incorrect.

Reference: "Parses a JSON string into a Python dictionary."
Generated: "Converts JSON text into a dict object."
Score: 5/5
Justification: Same meaning; different wording.

Reference: "Computes factorial of n using recursion."
Generated: "Computes factorial of n."
Score: 4/5
Justification: Correct but missing recursion detail.

---

Now evaluate the following method:

Reference: "{ref_summary}"
Generated: "{gen_summary}"

Output strictly in this format (start with 'Score: X/5'):
Score: X/5
Justification: <one or two sentences>
```

Figure 8: Prompt for evaluation of generated summaries against gold references.

## A.2 PROMPT FOR OUTPUT PREDICTION

```
You are given a full Python class and a unittest test case that
calls its methods.
Your task is to simulate the program step by step and predict
the exact output of the test method.
MUST think step by step and provide rationale for each step.
All the rationale must go inside the <think> tag and must NOT
mention the final output.

Complete it with the predicted value of the LEFT side of the
assertion (the exact output), and fill it in on the RIGHT side
of the assertion, keeping the same provided format of the
assertion and only replacing ?? with the predicted output.
Do NOT output anything extra.
Provide the completed assertion inside <answer> and </answer>
tags.


{code}

assert {test} == ??


<think>
Simulate the execution of the class and the unittest code,
tracking changes to operands, operators, and attributes.
Provide the rationale for each step, leading to the final
computed output.
</think>

<answer>
assert {test} == ??
</answer>

<think>
```

Figure 9: Prompt for ClassEval output prediction with unittest simulation.

## A.3 ADDITIONAL QUALITATIVE EXAMPLES

### A.3.1 CODE SUMMARIZATION

Figure 11 shows two LIVECODEBENCH cases where summaries remain correct under obfuscation, consistently capturing algorithmic intent. Unlike CLASSEVAL, we observe no regression into line-by-line narrations.

```
You are given a Python function and an assertion containing an input to
the function. Complete the assertion with a literal (no unsimplified
expressions, no function calls) containing the output when executing
the provided code on the given input, even if the function is incorrect
or incomplete. Do NOT output any extra information. Execute the program
step by step before arriving at an answer, and provide the full
assertion with the correct output in <answer> and </answer> tags,
following the examples.


def performOperation(s):
    s = s + s
    return "b" + s + "a"
assert performOperation(s = "hi") == ??

<think>
Let's execute the code step by step:

1. The function performOperation is defined, which takes a single
argument s.
2. The function is called with the argument "hi", so within the
function, s is initially "hi".
3. Inside the function, s is concatenated with itself, so s becomes
"hihi".
4. The function then returns a new string that starts with "b",
followed by the value of s (which is now "hihi"), and ends with "a".
5. The return value of the function is therefore "bhihia".
</think>
<answer>
assert performOperation(s = "hi") == "bhihia"
</answer>


{code}
assert {test} == ??


<think>
```

Figure 10: Prompt for LiveCodeBench output prediction with function execution.

### A.3.2 OUTPUT PREDICTION

Figure 12 illustrates a representative case from LIVECODEBENCH. Both original and obfuscated code yield the same reasoning trace and correct output, showing that identifier changes do not affect execution prediction.

### A.4 USAGE OF LLM ASSISTANCE

We used large language models (LLMs) to aid in polishing the writing and improving clarity. All research ideas, experiments, and conclusions are the work of the authors.

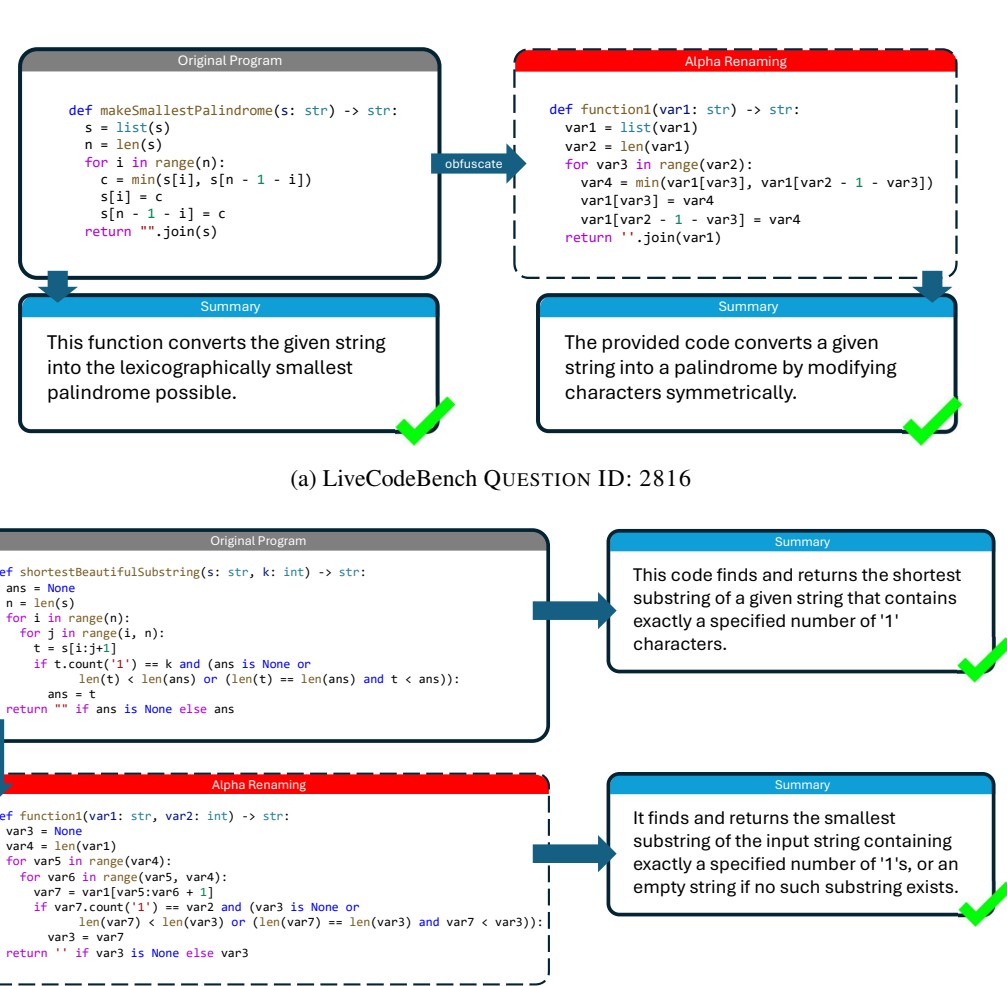

(a) LiveCodeBench QUESTION ID: 2816

(b) LiveCodeBench QUESTION ID: 3150

Figure 11: LIVECODEBENCH: consistent summarization under obfuscation. Both cases show that summaries capture algorithmic intent and remain stable despite identifier changes.

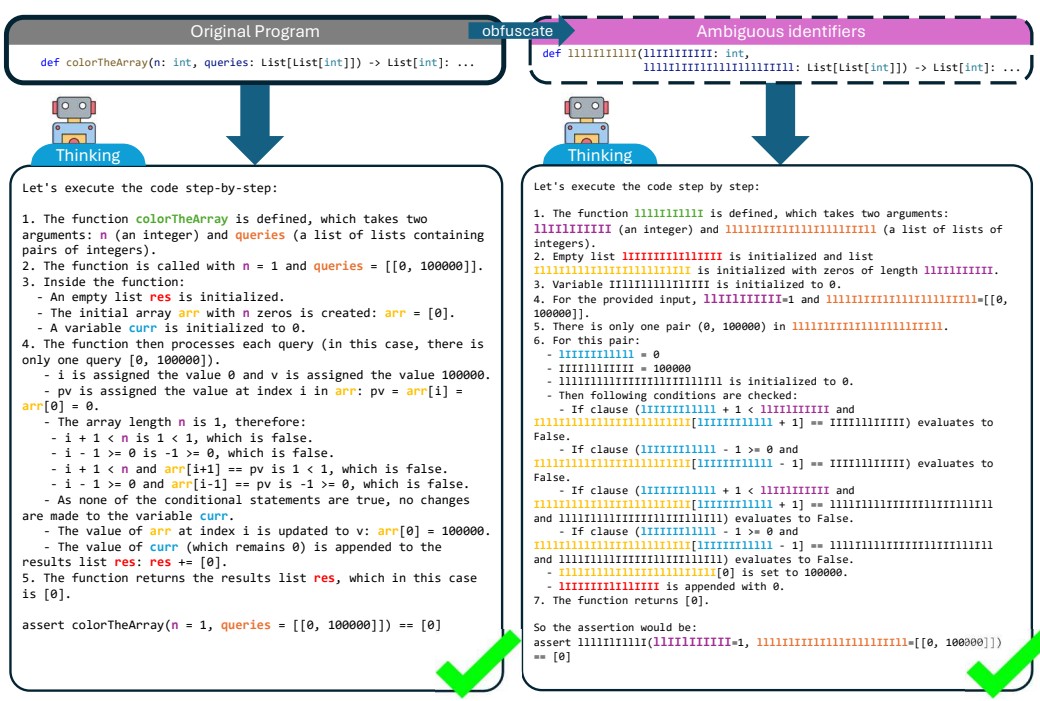

Figure 12: LIVECODEBENCH: stable reasoning and output across obfuscation.

