# OpenReview forum: "When Names Disappear: Revealing What LLMs Actually Understand About Code"
_ICLR.cc/2026/Conference — ICLR 2026 Conference Withdrawn Submission_

### Official Review · Reviewer_csYZ · 2025-10-19

**Soundness:** 2
**Presentation:** 2
**Contribution:** 1
**Rating:** 2
**Confidence:** 5

**Summary:**

This paper separately evaluates LLMs on code understanding with two channels: the formal structure and the human-readable names (intent). To achieve this, the paper applies various semantic-preserving obfuscations, such as renaming variables with meaningless or misleading terms, and evaluates LLMs on two tasks: code summarization and execution prediction.

The results show that removing meaningful names leads to notable performance drop on both tasks, highlighting that LLMs often rely on identifier names to understand code, instead of performing actual semantic reasoning, and this issue cannot be revealed with current benchmarks. Therefore, the paper releases its benchmark with obfuscation as a more robust way to evaluate LLMs' capabilities on code understanding.

**Strengths:**

- **Methodology:** The idea of two-channel evaluation makes sense, as ideally we want an LLM that can robustly understand code behaviors without the reliance on human-readable identifier names which could be misleading and ambiguous. This paper tries to quantify such robustness.
- **Experimental Design:** The evaluation includes code understanding at two different granularity levels, the high-level intent and the low-level execution. The evaluation samples also cover two categories, one is about competitive programming and the other is not, showing differences between them.
- **Empirical Results:** The paper presents several empirical results, including: (1) obfuscation leads to worse performance across various settings, (2) larger models are more robust. These results raise notable concerns about the true reasoning capability of LLMs.

**Weaknesses:**

- **Limited Novelty:** The community already acknowledges the lack of deeper code semantics understanding of current code LLMs, and using obfuscation to stress-test code LLMs is not new (e.g. [NIKIEMA et al. 2025](), [Fang et al. 2024](https://www.usenix.org/system/files/usenixsecurity24-fang.pdf)). Previous works also explore obfuscation beyond renaming variable names, such as refactoring control flows. All these papers show similar results, and I do not see enough new insights in this paper.
- **Insufficient Evaluation Efforts:** There is no evaluation on SOTA reasoning models, which can use more CoT steps to conduct more complicated analysis.
- **No Practical Guidance on Improvements:** This paper does not propose potential approaches to improve the robustness of code semantics understanding. Actually, the relatively small performance gap of code summarization on LiveCodeBench implies that a simple data-driven approach is likely to address the problem. That is, if we add more training data with obfuscated variable names, then the model will be more robust on such obfuscation-based evaluations. In fact, there are already some works proving this ([Paul et al. 2025](https://openreview.net/forum?id=VYvxrD7aS0)). Therefore, the evaluation proposed here does not point out a novel and challenging enough problem that the community has difficulty with, and provides limited new practical guidance on how to build better code LLMs for real-world coding tasks.

**Questions:**

Please see the section of weaknesses above, where the concerns are stated.

---

> ### Author Response · Authors · 2025-11-21
>
> Dear reviewer csYZ,
>
> Thank you for your thoughtful and constructive review. We appreciate the time and effort you invested, and we have carefully considered your comments. Below we provide point-by-point responses.
>
> ### **Q1 *[On Novelty Relative to Prior Obfuscation Work]***
> Thank you for raising this point. Prior works such as Fang et al. (2024) and Nikiema et al. (2025) indeed evaluate LLM robustness under a wide set of obfuscation operations, including dead-code insertion, control-flow rewriting, integer/string encryption, and other heavy syntactic transformations. Their goal is to study general degradation under complex obfuscation. Our contribution differs in two key ways.
>
> (1) Our work is intentionally different in scope and purpose: we focus only on **name-renaming**, because our aim is to isolate and characterize the **naturalness (naming) channel** of code, separate from structural reasoning. To study this cleanly, we pair name obfuscation with two complementary tasks that map directly onto the two channels: code summarization (intent channel) and execution prediction (structural channel). This targeted design leads to two insights that prior work does not highlight:
> + **Code Summarization task depends strongly on naming richness**, which explains why ClassEval and LiveCodeBench behave so differently. From that, intent-level evaluation like Code Summarization task should preferentially use datasets with rich, meaningful identifier names, such as ClassEval, rather than competitive-programming datasets like LiveCodeBench.
> + **Current execution benchmarks contain identifier leakage**, revealing that high scores often arise from name-based memorization rather than semantic reasoning, a form of benchmark inflation not isolated in earlier work.
>
> (2) Building on these findings, we propose a **practical augmentation protocol** that strengthens existing benchmarks by removing naming-based leakage without creating new datasets from scratch. The protocol has two steps: (i) generate fresh I/O pairs to break memorized execution mappings, and (ii) obfuscate identifiers to remove semantic cues from names. We then release **CLASSEVAL-OBF**, which applies this protocol systematically to provide a more reliable evaluation of LLMs. The key novelty is not in showing that obfuscation degrades performance, which is well known, but in demonstrating *why* current benchmarks are inflated by naming cues and offering a concrete, generalizable way to correct this.
>
> ### **Q2 *[On evaluating SOTA reasoning models with CoT]***
> We agree that reasoning depth is important, and our evaluation already incorporates this. All models we use are strong reasoning-capable LLMs, and our execution-prediction prompts explicitly include CoT guidance. As shown in Figures 9 and Figure 10 in Appendix (please refer to these figure), the prompt structure contains a `<think>` tag and a one-shot example that demonstrates step-by-step reasoning before producing the final answer. This ensures that each model is encouraged to perform multi-step analysis rather than shallow pattern matching.
>
>
> ### **Q3 *[On Practical Guidance for Improving Robustness]***
> We agree with the reviewer that data-driven approaches, such as training on obfuscated code, can improve robustness, as shown by Paul et al. (2025). Our focus in this paper is on revealing that current benchmarks inflate performance due to naming shortcuts memorization, which must be addressed before meaningful improvements can be measured. We view obfuscation-based training (including fine-tuning with behavior-correct outputs) as a natural next step, and we will add this direction as part of the future work enabled by our leakage-resistant evaluation protocol.
>
>
>
>
> We thank the reviewer again for the helpful feedback and look forward to discussing these points further.

---

### Official Review · Reviewer_ZFEH · 2025-11-01

**Soundness:** 3
**Presentation:** 3
**Contribution:** 3
**Rating:** 4
**Confidence:** 4

**Summary:**

This paper examines what LLMs truly “understand” about code by separating a structural semantic channel from a human naturalness channel. The authors evaluate models with a suite of semantics preserving obfuscations that progressively suppress identifiers and prose while keeping behavior unchanged. Across intent summarization and execution prediction, and across real world and competitive programming data, they find a consistent graded pattern: intent level performance declines sharply as naturalness is removed, while behavior level metrics remain largely stable except when names carry semantic content. The results offer converging empirical support for a two channel account of code understanding and motivate reporting before and after obfuscation deltas alongside human aligned intent metrics.

**Strengths:**

1. The paper’s starting point is reasonable. Constructing harder settings to test model robustness and generalization is necessary and can, to some extent, reflect a model’s generalization and resilience to misleading cues.
2. The work compares against cutting edge models, and the experimental results support the conclusions.

**Weaknesses:**

1. Does not assess whether the new evaluation correlates with scaling as model size increases.
2. The new evaluation introduces challenges, but can these be addressed quickly and easily with synthetic data? If a bit of synthetic data can solve it, the significance of this setting would be greatly reduced.
3. How to demonstrate that this evaluation helps real world coding tasks? After all, the real world does not replace variable names wholesale with meaningless symbols.

**Questions:**

please refer to weakness

---

> ### Author Response · Authors · 2025-11-21
>
> Dear reviewer ZFEH,
>
> Thank you for your thoughtful and constructive review. We appreciate the time and effort you invested, and we have carefully considered your comments. Below we provide point-by-point responses.
>
> ### **Q1 *[Correlation with model scaling]***
> In our experiments, we selected the strongest publicly available versions of each model because meaningful obfuscation effects only appear once the model already performs well on the original benchmark. Our goal is to observe whether models still perform well under obfuscation or whether their performance drops sharply, which reveals reliance on naming shortcuts. Smaller models often fail even on the non-obfuscated tasks, so the “gap” introduced by obfuscation becomes uninterpretable, they do not understand the code in the original setting, maybe model luckyly got correct answer with obfuscated code but not in original version, so scaling relationships cannot be meaningfully assessed. That said, applying our protocol to a broader scaling sweep is a natural extension, and we will include these experiments in future work.
>
> ### **Q2 *[Could synthetic data easily solve the new challenges?]***
> Our setting is not primarily about adding synthetic data, but about **fixing leakage in existing benchmarks**. The proposed protocol has two components:
> (1) generating new I/O pairs to break memorized mappings, and
> (2) obfuscating identifiers to remove naming shortcuts.
> This allows current benchmarks to be reused while providing a far more reliable signal of true code understanding. Even if synthetic data helps address parts of the problem, the central issue remains: **as long as benchmarks contain name-based leakage, reported performance will continue to misrepresent actual reasoning ability**. Our method targets the root cause rather than merely adding more data.
>
> ### **Q3 *[Relevance to real-world coding tasks]***
> The purpose of our work is to show that many widely used benchmarks overestimate model capability because models exploit name-based shortcuts; we do not claim that real-world code removes names. Obfuscation serves purely as a diagnostic tool to reveal this hidden dependence, not as a deployment scenario. Still, the findings are relevant for practical coding tasks: models perform poorly on obfuscated or unconventional code largely because such patterns are rare in training data and models have little reasoning experience with them. A straightforward way to improve robustness is to fine-tune on obfuscated code so the model learns to rely more on structure than on naming cues. Rejection sampling can further help: for each obfuscated input, keeping only behavior-correct outputs encourages the model to internalize structure-based reasoning rather than surface-level heuristics.
>
> We look forward to discussing these directions with you further.

---

### Official Review · Reviewer_yH4D · 2025-11-02

**Soundness:** 1
**Presentation:** 3
**Contribution:** 1
**Rating:** 2
**Confidence:** 5

**Summary:**

This paper investigates the extent to which Large Language Models (LLMs) rely on human-interpretable identifier names versus structural semantics to understand code. The authors argue that code communicates through 2 distinct channels:

(1) A formal, structural channel that entails formal semantics

(2) A "naturalness" channel that conveys intent through naming and other linguistic cues.

To disentangle these two channels, the authors introduce a suite of semantics-preserving obfuscation techniques that systematically degrade or remove identifier names while leaving the program’s underlying logic intact. They evaluate the performance of several state-of-the-art LLMs on 2 complementary tasks: code summarization and code execution prediction, using both their original and obfuscated versions. The results show that on intent-rich code, summarization performance collapses when names are removed, with models regressing to simple line-by-line descriptions. More surprisingly, the authors also observe a consistent performance drop on execution prediction tasks, which should theoretically only depend on the code’s structure. This suggests that current benchmarks may inadvertently reward memorization of naming patterns rather than genuine semantic reasoning.

**Strengths:**

## Novelty

**1. Comprehensive Obfuscation Suite**

While the core technique of identifier obfuscation is not new, this paper stands out for its systematic and multi-faceted approach. The introduction of a *suite* of semantics-preserving obfuscations, ranging from simple alpha-renaming to the use of ambiguous identifiers, cross-domain terms, and misleading semantics, provides a more granular and insightful analysis than previous studies that have typically focused on a single type of perturbation.

## Soundness

**1. Complementary Dataset Choices**

The authors' choice to evaluate models on both intent-rich, real-world code (ClassEval) and algorithmic, competitive-programming code (LiveCodeBench) is a key strength of the study. This allows them to demonstrate that the impact of naming is not uniform across all types of code, but rather depends on the extent to which the code relies on the "naturalness" channel for conveying intent.

**Weaknesses:**

## Novelty

**1. Insufficient Differentiation from Extensive Prior Work**

The paper presents variable name obfuscation as a novel framework for evaluating LLM code understanding, but this approach has been extensively explored in prior work. Gao et al. (2023) introduced the CREAM framework, which uses counterfactual reasoning to separate helpful from misleading identifier signals and explicitly employs identifier renaming to test model robustness [1]. Wang et al. (2022) developed ReCode, a comprehensive robustness evaluation benchmark that includes multiple schemes of variable renaming as semantics-preserving transformations [2]. Lam et al. (2025) proposed CodeCrash, which applies systematic renaming among other logic-preserving perturbations to expose over-reliance on natural language cues [3]. Even earlier, Rabin et al. (2021) studied the generalizability of neural program models with respect to semantic-preserving transformations, including identifier renaming [4]. The core technique of using identifier obfuscation to test semantic understanding is well-established in the literature. While the paper mentions CREAM and CodeCrash, it does not provide a detailed comparison to clearly delineate what is genuinely novel beyond applying an existing technique to a slightly different set of tasks and models.

**Suggestion:** It would be better to include a dedicated subsection that explicitly compares the proposed approach with CREAM, ReCode, CodeCrash, and other prior work on semantic-preserving transformations. The authors should clearly articulate what specific methodological innovations or insights their work provides beyond the existing literature, rather than presenting identifier obfuscation as if it were a new concept.

## Soundness

**1. Fundamental Flaw in Summarization Task**

The paper's use of code summarization as a proxy for "intent-level understanding" suffers from a critical methodological flaw that undermines the validity of the conclusions. The ground truth summaries almost certainly contain information that can only be inferred from variable names, not from program structure alone. For example, as illustrated in Figure 1, after name obfuscation, there is no way to determine that the original program implements a "Minesweeper game" rather than any other grid-based application with identical control flow—it could equally represent a spreadsheet calculator, a Conway's Game of Life implementation, or a warehouse inventory system. Similarly, a simple multiplication function `result = a * b` could represent price × quantity, discount × price, or units × count—all implemented identically but with entirely different semantic meanings conveyed solely through naming. When the ground truth summary includes domain-specific concepts like "Minesweeper" or "pricing calculation," the model is being evaluated on information that has been deliberately removed from the input. This creates an inherently unsolvable task after obfuscation, making the performance drop inevitable and uninformative about the model's structural reasoning capabilities. The evaluation would only be sound if the ground truth summaries were verified to be abstract enough that they contain no information implied by variable naming—a condition the paper does not establish or enforce.

**Suggestion:** The authors should either (1) create a new set of ground truth summaries that are provably derivable from structure alone, excluding any domain-specific or application-level concepts that require naming cues, or (2) acknowledge this as a fundamental limitation and reframe the summarization results as measuring "naming-dependent intent recovery" rather than "structural understanding." Additionally, a human evaluation study could assess whether the obfuscated summaries produced by models are actually incorrect or simply more abstract than the naming-rich ground truth, which would help clarify whether the performance drop reflects a genuine failure or an artifact of the evaluation design.


**2. Incomplete Analysis of Execution Prediction Results**

While the paper shows that execution prediction accuracy drops under obfuscation, this finding does not conclusively demonstrate a fundamental flaw in the models' understanding of program semantics. An alternative and equally plausible interpretation is that descriptive variable names simply provide shortcuts that make correct predictions easier to generate on the first attempt, without implying that the model lacks the capability to reason through the obfuscated version given more attempts. If the authors had evaluated Pass@5, Pass@10, or higher values of k instead of only Pass@1 and Pass@3, it is quite possible that the accuracy on obfuscated programs would converge to that of original programs. Such convergence would suggest that naming provides efficiency gains (reducing the number of attempts needed to find the correct answer) rather than revealing a fundamental reasoning deficit. The current experimental design does not rule out this possibility, which significantly weakens the claim that models are "memorizing naming patterns rather than genuinely reasoning about semantics." Without exploring higher values of k or analyzing the distribution of correct answers across multiple samples, the interpretation remains ambiguous and the conclusions overstated.

**Suggestion:** It would be worthwhile to extend the evaluation to Pass@5, Pass@10, or even Pass@20 to investigate whether performance on obfuscated code converges to that of the original code. Additionally, analyzing the variance and consistency of predictions across multiple samples could help distinguish between "naming as a shortcut" and "naming as a prerequisite for reasoning." If convergence is observed, the authors should revise their claims to acknowledge that the results may reflect efficiency differences rather than fundamental capability gaps. This would provide a more nuanced and accurate understanding of the role of identifiers in execution prediction.

**3. Limited Model Diversity Undermines Generalizability**

The paper evaluates only four models (GPT-4o, Qwen3-Coder 480B, DeepSeek V3, Llama 4 Maverick), which is insufficient to support broad claims about how LLMs understand code. To establish that the observed phenomena are not artifacts of specific training procedures or architectures, the evaluation should include systematic variation across multiple dimensions. Specifically, the study lacks: (1) multiple models from the same family at different scales (e.g., 7B, 13B, 70B versions of Llama or Qwen) to assess whether the reliance on naming is scale-dependent, and (2) explicit comparison between reasoning models versus standard models (Qwen 3 in thinking mode v.s. non-thinking mode) from the same family to determine whether reasoning capabilities mitigate the naming dependency. Without this systematic exploration, it remains unclear whether the findings generalize across the broader landscape of code LLMs or are specific to the particular models chosen.



## Significance

**1. Underexplored Practical Implications**

The paper's findings have important implications for the development and evaluation of code LLMs, but the discussion of these implications remains somewhat abstract. The paper concludes that its new benchmark will provide a more reliable basis for assessment, but it could offer more specific, actionable recommendations. For example, how should developers change their coding practices in light of these findings? Should they prioritize more descriptive names, or should they focus on writing code with clearer structure? How can LLM developers use these insights to build more robust models—should they augment training data with obfuscated code, modify the training objective, or employ different architectural choices? The lack of concrete guidance limits the practical impact of the work.

**Suggestion:** It might be worthwhile to add a discussion section that explores the practical implications of the findings in more detail. This could include concrete recommendations for developers on how to write code that is more amenable to LLM analysis (or conversely, how to recognize when LLMs may struggle) and for researchers on specific training strategies or architectural modifications that could help LLMs become more robust to identifier obfuscation while maintaining high performance on naturally-named code.


## Effectiveness

**1. Unvalidated Real-World Utility of the Proposed Benchmark**

The paper convincingly shows that CLASSEVAL-OBF can mitigate the problem of identifier leakage and provide a more accurate measure of a model's reasoning ability in a controlled experimental setting. However, the ultimate goal of a benchmark is to predict a model's performance on real-world tasks. The paper does not provide any evidence to show that performance on CLASSEVAL-OBF is a better predictor of downstream task performance (such as bug fixing, code generation in real-world software projects, or code review) than existing benchmarks. Without this validation, it remains unclear whether optimizing for CLASSEVAL-OBF performance would actually lead to models that are more useful in practice, or whether it might inadvertently optimize for a different set of capabilities that are less relevant to real-world applications.

**Suggestion:** It would be better to include an experiment that investigates the correlation between performance on CLASSEVAL-OBF and performance on downstream tasks such as bug fixing, code generation in real-world software projects, or repository-level code understanding. This would provide stronger evidence for the practical utility of the proposed benchmark and help establish that it measures capabilities that matter for real-world applications.



## References

[1] [Two sides of the same coin: Exploiting the impact of identifiers in neural code comprehension (Gao et al., 2023)](https://ieeexplore.ieee.org/document/10172869/)

[2] [ReCode: Robustness Evaluation of Code Generation Models (Wang et al., 2022)](https://arxiv.org/abs/2212.10264)

[3] [CodeCrash: Stress Testing LLM Reasoning under Structural and Semantic Perturbations (Lam et al., 2025)](https://arxiv.org/abs/2504.14119)

[4] [On the generalizability of Neural Program Models with respect to semantic-preserving program transformations (Rabin et al., 2021)](https://www.sciencedirect.com/science/article/pii/S0950584921000379)

**Questions:**

## Clarification Questions

1. In Section 5.1, you describe four obfuscation strategies. Could you provide more detail on how the "cross-domain terms" and "misleading semantics" were generated? Was this done manually or automatically? If automatically, what was the process?

2. The LLM-as-a-judge evaluation for summarization is based on a rubric from the BigGenBench framework. Could you elaborate on the specific criteria used in this rubric and how they were weighted to produce the final scores?


## Discussion Questions

1. Given the fundamental tension between obfuscating names and evaluating against naming-dependent ground truth summaries, do you believe the summarization task can be salvaged (e.g., by creating structure-only ground truth), or should it be replaced with a different task that is provably solvable from structure alone?

2. If Pass@10 or Pass@20 on obfuscated code converges to the performance on original code, would you still interpret the results as indicating a fundamental flaw in LLM reasoning, or would you agree that naming primarily provides efficiency shortcuts?

3. Your work focuses on analyzing the understanding of existing code. How do you think these findings might apply to the task of code *generation*? Would a model that is less reliant on naming cues be better at generating novel code that solves new problems, or might it struggle to produce code that is readable and maintainable?

4. What specific training strategies or architectural modifications do you think could help LLMs become more robust to identifier obfuscation while maintaining high performance on naturally-named code? For example, should training data be augmented with obfuscated examples, or should the model architecture be modified to better capture structural semantics?

---

> ### Author Response · Authors · 2025-11-21
>
> Dear reviewer yH4D,
>
> Thank you for your thoughtful and constructive review. We appreciate the time and effort you invested, and we have carefully considered your comments. Below we provide point-by-point responses.
>
> ## Clarification Questions
>
> ### **Q1 *[More detail on obfuscation method implementation]***
> Both cross-domain and misleading-semantics names are generated fully automatically from the original identifier and an obfuscation seed. For cross-domain terms, we hash each identifier and use the digest to select one or two tokens from a fixed medical-terminology vocabulary, then append a short hex suffix to ensure determinism and uniqueness. Misleading-semantics names follow the same hashing pipeline but route each identifier into class-, method-, or variable-specific pools containing realistic yet intentionally deceptive names (e.g., *DatabaseConnection*, *compute_max*). Collisions are handled programmatically with numeric suffixes. No manual intervention is involved; all replacements are deterministic and semantics-preserving.
>
> ### **Q2 *[Clarification on LLM-as-a-judge rubric]***
> Please see *Figure 8* in the Appendix, which contains the full evaluation prompt and rubric.
>
> ## Discussion Questions
>
> ### **Q1 *[Naming-dependent ground truth vs. obfuscation]***
> Summarization is intentionally used because it evaluates the *intent* channel, where names naturally carry semantic information. If we replaced it with a structure-only task, we would no longer be measuring intent understanding but instead the structural channel, which we already analyze in Section 5 through execution prediction. Thus, the goal is not to “salvage” summarization under obfuscation, but to show that removing names exposes how much summarization performance depends on naming cues, precisely the phenomenon our study aims to measure.
>
> ### **Q2 *[Interpreting Pass@10 or Pass@20 convergence]***
> If Pass@10 or Pass@20 on obfuscated code converges to the performance on original code, then yes, this would suggest that naming does not fundamentally affect the model’s *ability* to solve structure-only tasks when given enough samples. In other words, names act more as efficiency shortcuts rather than altering the reachable solution space. However, practical coding scenarios and standard benchmark protocols rely on low-pass settings (Pass@1 or Pass@3), which reflect the model’s real behavior under single-attempt or few-shot constraints. Our conclusions therefore focus on these realistic settings, where obfuscation consistently causes substantial degradation so we not run on high pass rate like that (Pass@10 and Pass@20).
>
> ### **Q3 *[Implications for code generation]***
> Our findings extend to code generation as well. In additional experiments (not included in the paper), we asked models to generate code using intentionally meaningless or obfuscated names. Their performance dropped substantially compared to normal generation. This suggests that current models rely heavily on familiar naming patterns from training data, and struggle when those cues are removed, consistent with our analysis of code understanding. A model less name-dependent could generalize better to novel tasks, but may also need explicit training objectives to maintain readability and maintainability.
>
> ### **Q4 *[Training strategies for robustness to obfuscation]***
> A straightforward direction is to fine-tune on obfuscated code so the model learns to rely on structure rather than names, especially since current models see almost no obfuscated code during training and therefore hallucinate when faced with these unfamiliar patterns. “Rejection sampling” can further help: for each obfuscated input, we keep only outputs that correctly reflect program behavior, pushing the model toward structure-based reasoning. Although our paper focuses on using obfuscation to reveal benchmark leakage, rather than proposing a full training solution, we appreciate the reviewer highlighting this direction and plan to explore such fine-tuning approaches in future work.

---

### Official Review · Reviewer_GDDB · 2025-11-03

**Soundness:** 2
**Presentation:** 3
**Contribution:** 3
**Rating:** 4
**Confidence:** 4

**Summary:**

The paper introduces a code understanding benchmark to probe the semantic understanding of Code LMs by subjecting it to summarization and output prediction tasks on obfuscated code generated from well-known benchmarks.

**Strengths:**

1. The authors have explored a taxonomy of obfuscation types
2. The obfuscation codes are sourced from code snippets that are solutions to relevant benchmarks that are not too easy and also commonly used for evaluating code LMs.

**Weaknesses:**

1. Some interesting ablations for gauging the effect of obfuscation on Code LMs is missing, which could enhance the paper:

1.1 The authors could have explored the effect of obfuscating various types of identifiers separately e.g. class names, function names, variables, etc
1.2 The authors should look into the effect of obfuscating varing proportions of identifiers. It would be interesting to note at what points the degradtions of code understanding set in.

2. Some very relevant existing work on training model's to be robust to these perturbations [1,2,3] is missing from the literature and discussions.

[1]	Marie-Anne Lachaux, Baptiste Rozière, Marc Szafraniec, Guillaume Lample: DOBF: A Deobfuscation Pre-Training Objective for Programming Languages. NeurIPS 2021: 14967-14979

[2] 	Indraneil Paul, Haoyi Yang, Goran Glavas, Kristian Kersting, Iryna Gurevych: ObscuraCoder: Powering Efficient Code LM Pre-Training Via Obfuscation Grounding. ICLR 2025

[3].   Seyedreza Mohseni, Seyedali Mohammadi, Deepa Tilwani, Yash Saxena, Gerald Ketu Ndawula, Sriram Vema, Edward Raff, Manas Gaur: Can LLMs Obfuscate Code? A Systematic Analysis of Large Language Models into Assembly Code Obfuscation. AAAI 2025: 24893-24901

**Questions:**

N/A

---

> ### Author Response · Authors · 2025-11-21
>
> Dear reviewer GDDB,
>
> Thank you for your thoughtful and constructive review. We appreciate the time and effort you invested, and we have carefully considered your comments. Below we provide point-by-point responses.
>
> ### **Q1 *[Ablations on identifier types and obfuscation proportion]***
>
> **Q1.1** We appreciate the suggestion to analyze obfuscation by identifier category (e.g., classes, methods, variables). We agree this is a valuable direction and would provide finer-grained insight into where degradation originates. In this submission, we focused on *full* semantics-preserving obfuscation because our primary goal is to evaluate **benchmark reliability under the strongest form of name suppression**, where identifier leakage and memorization shortcuts are most clearly exposed. Isolating specific identifier types is therefore orthogonal to our core objective but nevertheless informative. We will include type-specific ablations in the revised version to strengthen the completeness of our analysis.
>
> **Q1.2** We concur that varying the *proportion* of identifiers obfuscated is an interesting extension. Our experiments used full obfuscation because partial obfuscation can leave semantic anchors intact, allowing models to still leverage memorized patterns, making it harder to diagnose evaluation leakage. However, a proportional ablation curve would indeed help characterize *when* degradation sets in and how sharply performance declines. We will include this analysis in the revised version to provide a more comprehensive understanding of the degradation dynamics.
>
> ### **Q2 *[Missing related work on robustness-oriented obfuscation training]***
> Thank you for highlighting DOBF, ObscuraCoder, and Mohseni et al.; we will incorporate these references into the Related Work section. These methods primarily employ obfuscation as a **training signal**, for deobfuscation objectives, obfuscation-grounded pretraining, or generating obfuscated code - aimed at improving model robustness. Our contribution is complementary but distinct: we use obfuscation **purely as an evaluation tool** to uncover identifier leakage and memorization shortcuts in existing benchmarks. Our results show that once naming cues and original I/O mappings are removed, performance drops significantly even on execution tasks, indicating that current benchmarks may overestimate semantic reasoning. CLASSEVAL-OBF and our I/O-augmentation protocol therefore aim to provide **leakage-resistant, semantics-grounded evaluation**, rather than proposing an additional robustness-oriented training approach.

---

### Note · Authors · 2025-12-30

I have read and agree with the venue's withdrawal policy on behalf of myself and my co-authors.